# Integration of Deep Learning and Active Shape Models for More Accurate Prostate Segmentation in 3D MR Images

**DOI:** 10.3390/jimaging8050133

**Published:** 2022-05-11

**Authors:** Massimo Salvi, Bruno De Santi, Bianca Pop, Martino Bosco, Valentina Giannini, Daniele Regge, Filippo Molinari, Kristen M. Meiburger

**Affiliations:** 1Biolab, PolitoBIOMed Lab, Department of Electronics and Telecommunications, Politecnico di Torino, Corso Duca degli Abruzzi 24, 10129 Turin, Italy; massimo.salvi@polito.it (M.S.); bianca.pop51@gmail.com (B.P.); filippo.molinari@polito.it (F.M.); 2Multi-Modality Medical Imaging (M3I), Technical Medical Centre, University of Twente, PB217, 7500 AE Enschede, The Netherlands; b.desanti@utwente.nl; 3Department of Pathology, Ospedale Michele e Pietro Ferrero, 12060 Verduno, Italy; mbosco@aslcn2.it; 4Department of Surgical Sciences, University of Turin, 10126 Turin, Italy; valentina.giannini@unito.it (V.G.); daniele.regge@unito.it (D.R.); 5Department of Radiology, Candiolo Cancer Institute, FPO-IRCCS, 10060 Candiolo, Italy

**Keywords:** active shape models, automatic prostate segmentation, convolutional neural network, deep learning, hybrid framework, medical image segmentation, MRI

## Abstract

Magnetic resonance imaging (MRI) has a growing role in the clinical workup of prostate cancer. However, manual three-dimensional (3D) segmentation of the prostate is a laborious and time-consuming task. In this scenario, the use of automated algorithms for prostate segmentation allows us to bypass the huge workload of physicians. In this work, we propose a fully automated hybrid approach for prostate gland segmentation in MR images using an initial segmentation of prostate volumes using a custom-made 3D deep network (VNet-T2), followed by refinement using an Active Shape Model (ASM). While the deep network focuses on three-dimensional spatial coherence of the shape, the ASM relies on local image information and this joint effort allows for improved segmentation of the organ contours. Our method is developed and tested on a dataset composed of T2-weighted (T2w) MRI prostatic volumes of 60 male patients. In the test set, the proposed method shows excellent segmentation performance, achieving a mean dice score and Hausdorff distance of 0.851 and 7.55 mm, respectively. In the future, this algorithm could serve as an enabling technology for the development of computer-aided systems for prostate cancer characterization in MR imaging.

## 1. Introduction

Prostate cancer is the most common cancer in the world for what concerns the male population [1]. There are many techniques that can be used for diagnosis, such as screening using the serum prostate specific antigen (PSA) or urological approaches, such as biopsies guided by transrectal ultrasound; however, the method that demonstrates the highest accuracy is MRI (magnetic resonance imaging). The use of MRI in prostate cancer diagnosis has the primary purpose of determining the stage of the disease, and potentially planning the necessary radiation therapy, both in form of external beam radiation, and in the form of a guide for the seed implant for brachytherapy.

Prostate gland (PG) boundary delineation represents an important role for radiation therapy planning, MRI-guided biopsy and development of computer-aided diagnosis systems for the early detection of prostate cancer [2]. However, manual segmentation of the prostate in 3D MRI images is time-consuming and prone to inter- and intra-operator variability. In this regard, fast and accurate automated segmentation algorithms are highly demanded in clinical practice. Automatic segmentation of the PG is a very challenging task for several reasons: (i) different MRI scans may have global interscan variability and intra-scan intensity variations; (ii) prostates exhibit a wide range of morphological and intensity variations among healthy subjects and is exacerbated in pathological conditions; (iii) PG is surrounded by anatomical structures with similar intensity distribution, such as vessels, bladder, rectum and seminal vesicles; (iv) image artifacts such as susceptibility artefacts and magnetic field inhomogeneities are often present. Moreover, the boundaries of the gland are not clearly defined, especially for the apex and base regions of the prostate, which are the most difficult to segment yet are of fundamental importance in clinical practice [2,3].

Active shape models (ASMs) have been developed to address the need to model structures that have great variability in size, shape, and appearance. These models consist of flexible sets of labeled points representing an object, from which neighboring point statistics are analyzed over several training forms using the Point Distribution Model (PDM) [4,5]. Traditional ASMs have problems segmenting the prostate in MRI images because of the large variability in contours between different patients. They are rarely used alone because they require handcrafted features to model the contour at the edges. In addition, contour segmentation is complicated by the fact that the shape initialization is often incorrect or not precise enough [6].

Nowadays, deep learning frameworks have become the main methodology for analyzing medical images [7,8,9]. Due to their powerful learning ability and advantages in dealing with complex patterns, deep learning algorithms are ideal for image analysis challenges, particularly in the field of MRI [10,11]. Yu et al. [12] presented a volumetric Convolutional Neural Network (CNN) to improve the training efficiency and the ability of the network to discriminate examples under limited training data. The volumetric network had a down-sampling path to extract image features and an upsampling path to generate the final prediction. Zhu et al. [13] implemented a 3D network inspired by UNet and DenseNet for domain image segmentation (SNet). Two SNets segment source and target domain images separately. A discriminator differentiates feature representations from the former or the latter in an adversarial fashion. Jia et al. [14] proposed a Generative Adversarial Network (GAN) to solve the problem of anisotropic spatial resolution of prostate MRI images. This GAN, called 3D APA-Net, consisted of two modules: the first was a 3D encoder–decoder segmentation network while the second module was a CNN discriminator used to refine the final result.

Recently, some hybrid approaches based on the complementary use of deep learning techniques and statistical models have been published to achieve higher performance in prostate segmentation. Cheng et al. [15] used a CNN-based architecture and a training strategy based on an active shape model to solve the problem of shape variability and the limited number of training data. Kamiri et al. [16] employed an active appearance model (AAM) as an initialization to obtain an estimate of prostate edges with MRI volumes. AAM alone produces edges that are not always accurate; the shapes obtained are often irregular and the segmentation results are unstable over the range of model search. Therefore, a CNN is applied to achieve segmentation refinement at prostate edges. He et al. [6] presented an approach that combines an adaptive feature learning boosting tree (AFL-PBT) with an active shape model. The AFL-PBT relies on both handcrafted features and features obtained through deep methods to pre-segment the prostate and create a searching band for the statistical model. Ushinsky et al. [17] presented a hybrid 3D-2D U-Net that leverage features from multiple axial slices simultaneously to better reconstruct a single 2D image. Meyer et al. [18] proposed an anisotropic 3D multi-stream CNN architecture, which processes additional scan directions to produce a high-resolution isotropic prostate segmentation. Pollastri et al. [19] designed a CNN-based 3D model to extract multiscale 3D features with dilated convolutions, taking advantage of a self-attention transformer to obtain a global field of view. Most of the above methods have been evaluated for comparison with a single manual reference segmentation. However, due to the large variability between operators, it is important to compare the results of the segmentation algorithm with more than one experienced operator. In addition, it is important to choose a set of metrics that is sensitive to different types of errors, such as surface disagreement, regional mismatch, and volume differences [20]. Table 1 summarizes all the approaches described in this section, along with the dataset used and the performance obtained. 

In this paper, a novel prostate gland segmentation strategy is presented. Our approach starts from an initial segmentation of the prostate using a volumetric CNN model to fully exploit the 3D spatial contextual information of the input data to perform a volume-to-volume prediction. The proposed deep network is three-dimensional, thus giving more importance to spatial coherence that allows obtaining the organ in its entirety, compared to a two-dimensional network. Although three-dimensional segmentation has this spatial advantage, it places less emphasis on accuracy on the individual slice, especially at the edges of the prostate [24]. For this reason, an improved active shape model is proposed to perform a refinement of the prostate contours on the CNN outputs to achieve a more precise segmentation and to adjust any implausible shapes output by the network as they are corrected according to the shape constraints defined by the ASM parameters. The main contribution of this paper can be summarized as follows:We propose a fully automated hybrid approach for 3D MRI prostate segmentation using CNNs and ASMs. A custom 3D deep network is employed to provide a first segmentation of the prostate glands. Then, a statistical model is applied to refine the segmentation in correspondence of the prostate edges.We develop an effective approach based on the combination of semantic features extracted from the CNN and statistical features from the ASM. By using a CNN-based initialization, we can bypass the limitations of current ASMs.We improved the robustness of the ASM by removing noisy intensity profiles using the DB-SCAN clustering algorithm.We publicly release the code used in this work. An extended validation is also performed by comparing the proposed approach with two manual operators. Our algorithm obtains highly satisfactory results.

## 2. Materials and Methods

### 2.1. Patients and Dataset Composition

The dataset is composed of T2w MRI prostatic volumes of 60 male patients. T2w imaging was employed since it provides good contrast between anatomical characteristics of the organ and for this reason are helpful in concluding a diagnosis [25]. The study was approved by the local Ethics Committee and all the participants signed an informed consent form (protocol number: CE IRCCS 191/2014, approval date: 17 June 2014). The images were collected with a 1.5 T scanner (Signa Excite HD, GE Healthcare, Chicago, IL, USA) using a four-channel phased-array coil combined with an endorectal coil (Medrad, Indianola, PA, USA). The following protocol was adopted to acquire the T2w images: slice thickness of 3 mm covering 7.2 cm, hence obtaining 24 slices per volume; field of view equal to 16 × 16 cm; acquisition matrix of 384 × 288 with a reconstruction matrix of 512 × 512 pixels. The prostate gland was segmented in T2w images by an expert radiologist, with 11 years of experience in MRI prostate examination. Manual annotations have been obtained using the 3D Slicer software [26]. An example of manual annotations is shown in Figure 1. Our dataset was divided into a construction set (45 patients) and test set (15 patients). The construction set was then divided into a training set (36 patients) and a validation set (9 patients).

### 2.2. Bias Field Correction and Intensity Normalization

Pre-processing is applied to improve the quality of the MR images by enhancing contrast and making the process of feature recognition easier for the algorithm. MRI images can be characterized by non-uniformity of intensity at low frequency that corrupts the images (i.e., bias field). Bias field correction is performed using the N4 algorithm [27] which adopts an iterative scheme to estimate and remove the distortion field. The algorithm usually takes an image as input and estimates its bias field using an automatically defined mask by applying Otsu’s threshold algorithm. Let us consider the MRI image defined as: (1)v(x)=u(x)f(x)+n(x)
where v is the image, u represents the uncorrupted image, f is the bias field and n is the Gaussian noise. The iterative procedure followed by the N4 algorithm allows us to obtain the uncorrupted image at the n^th^ iteration as: (2)$${\hat{u}}^{n}=\hat{v}-\overset{n}{\underset{i=1}{\sum }}f^{i}$$

The optimization is applied to the residual bias field, and the total bias field is obtained as the sum of the residuals. Compared to other techniques [28], the N4 algorithm does not use least squares fitting, and because of this, it does not need any regularization parameters to be adjusted, so it is more robust to noise because the approximation is done locally and then all the local solutions are combined to obtain the global approximation.

The lack of a standard image intensity scale in MRI makes both visualization and subsequent image analysis challenging. MRI intensities do not have a fixed meaning due to different protocols, patients, and scanners. This can cause various problems during image segmentation and quantification of radiomic features [29,30,31]. Therefore, we normalized the intensity of the MRI volumes according to the method proposed by Làszlò et al. [32]. Briefly, several reference points (i.e., percentile values, mode) are extracted from the grayscale histograms of the training set volumes to compute a standardized histogram. Then, for each MRI volume, image mapping is applied from the original histogram space to the standardized one through piecewise linear transformations. More details about the intensity normalization are provided in Appendix A. Figure 2 shows the pre-processing steps employed. 

### 2.3. Deep Convolutional Networks

A custom network, called VNet-T2, is designed to perform an initial segmentation of the prostate gland. The basic idea is to create a network inspired by the UNet architecture [33] that, taking as input a prostate volume, learns from it to discriminate between voxels belonging to the prostate and the background, and returns a volumetric label of the same size as the input. The network discussed in this section is trained from scratch since the weights obtained for similar problems employed different input sizes, and due to the complexity and high computational load of the volumetric problem, it was decided to avoid resampling to make the sizes compatible. This means that the original size of the volumes was maintained (512 × 512 × 24 pixels) and the input size of the network was consequently customized.

The VNet-T2 architecture, based on the 3D-UNet Keras model [34], is shown in Figure 3. This network has alternating blocks of 3D convolution, 3D max-pooling, activation, and 3D deconvolution. The VNet-T2 network consists of an initial deconvolution path composed of convolution and max pooling blocks to extract image features at deeper levels. Specifically, the individual blocks of the encoder have been structured as follows:A first convolutional layer (kernel size 3 × 3 × 3) followed by a layer of instance normalization;A second convolutional layer (kernel size 3 × 3 × 3) followed by a layer of instance normalization;A max-pooling layer in 3D with a pool size equal to (2, 2, 2).

The spatial information is reconstructed in the subsequent upsampling path composed of alternating blocks of transposition and convolution to restore the original dimensions. The output of the network is a probability map that assigns each voxel a probability of belonging to a specific class.

The entire network is trained on a two-class problem (prostate vs. background), giving the 512 × 512 × 24 grayscale images as input and the corresponding labeled masks as the target. To solve the problem of class imbalance, two different strategies have been tested:Class balancing: the network’s loss function is class-weighted by considering how frequently a class occurs in the training set. This means that the least represented class (prostate gland) will have a greater contribution than the more represented one (background) during the weight update. This is performed following the same approach of our previous work [35].Dice loss: the network loss function is calculated as 1-DSC, where DSC is the dice score computed between the manual annotation and the network prediction. The dice overlap is a widely used loss function for highly unbalanced segmentations [36].

To optimize all hyperparameters in the network, we tried different configurations, varying the network depth, number of filters, loss function, learning rate, and evaluation metric. The maximum number of epochs was set to 50, with a validation patience of 25 epochs for early stopping of the training process. The final configuration of the VNet-T2 network is summarized in Table 2. The best model was chosen by analyzing the performance obtained on the training set (36 patients). We have made the image dataset and implementation codes of the VNet-T2 network publicly available (see Data Availability Statement). Training was performed on a workstation with 32 GB of RAM with access to a Nvidia K80 GPU (Nvidia, Santa Clara, CA, USA) using Keras framework and TensorFlow as backend.

### 2.4. Active Shape Models (ASM)

The results obtained in output from the VNet-T2 network present a rough segmentation of the prostate that needs further refinement. To obtain the final precise segmentation of the prostate in the MRI volumes, an ASM approach was employed. 

The Point Distribution Model (PDM) [37] represents the modes of variation of the shape of a structure, having the average shape calculated on the training set as the initial model. From this average shape, the position, orientation, scale and shape parameters are adjusted to obtain a better fit of the image. Limits are placed on the movements that every point can make, to maintain the shape that can be obtained plausible (Global Shape Constraints) [4]. Hence, the first step in the ASM refinement is the construction of a mean shape model on the training dataset.

#### 2.4.1. Mean Shape Model Determination and Appearance Data

In order to create a mean shape model, a set of coherent vertices for all the prostates in the dataset must first be determined, after which the final shape model can be built. After an initial isotropic upsampling in order to have the same resolution (0.5 mm × 0.5 mm × 0.5 mm) along all three axes, the computation of the coherent vertices is divided into 4 steps:

1.3D-surface fit: in this step, the external surface of each prostate is fitted with a three-dimensional ellipsoidal surface. Subsequently, key reference points are determined in both the *x-y* plane and the *y-z* planes as $F_{xy}=\left[x_{center},y_{center}\pm \frac{c_{xy}}{2},z_{center}\right]$ and $F_{xz}=\left[x_{center}\pm \frac{c_{xz}}{2},y_{center},z_{center}\right]$, respectively, where $c_{xy}=\sqrt{radius_{max}^{2}-radius_{min}^{2}}$ and $c_{xz}=radius_{min}^{2}$.2.Triangulation: Starting from the 3D vertices obtained in step 1, the Alpha Shape Triangulation [38] is employed to divide the 3D surface into a variable number of triangles. This triangulation method requires an α parameter that defines the level of refinement of the structure. A value of α = 50 was used, as it was found to be appropriate for all prostate shapes included in the dataset.3.Ray-Triangle intersection: This step is necessary to obtain corresponding key points in each prostate. To do so, the Moller-Trumbdor [38] algorithm is employed to compute the intersection between a set of rays originating in each of the key points found in step 1 (i.e., *F_xy_* and *F_xz_*) and each of the triangles that were obtained in step 2. A ray is defined as R(t)=OP+tD where *O* is the origin of the ray and *D* is a normalized direction. In this study, we chose 8 directions with a step angle (θ) of 360°/8. For a detailed description of how this algorithm works, please see the study by Moller et al. [38].4.Vertices determination: For each direction *D*, the intersection points between the ray and the 3D model are determined and make up the final vertices.

For the final mean shape model determination, the vertices obtained in step 4 for all volumes in the training set (e.g., manual segmentations) need to be aligned. For simplicity, the contour positions of all volumes are aligned to the first volume in the dataset. Alignment is obtained by removing any potential translation (i.e., the volumes are centered) and by correcting the rotation using the four-quadrant inverse tangent in two steps (Figure 4). Subsequently, a matrix containing all the contour point data of the training set is constructed and a principal component analysis (PCA) is applied to remove contour noise by maintaining only 98% of the total variance, which produces the final mean shape model (Figure 4).

The next step in our ASM approach is the determination of the appearance data (i.e., gray levels) that need to be employed to optimize the finale prostate contour. A single-resolution model (scale = 1) based on the values of the gray profiles along the normal directions of the surface points is employed. The gray level intensity profiles along the normal directions of the surface are obtained by determining a length of the profile, $length=\left(n_{s}*2\right)+1$, which is centered on the surface and protrudes both within and outside the surface. In this study, $n_{s}$ is equal to 2. A cubic interpolation is then applied on the gray level intensity profiles and subsequently a clustering algorithm was employed. Since the prostate presents a great variability along the contours, the application of the density-based spatial clustering of applications with noise (DB-SCAN) algorithm [39,40] was crucial to find clusters of an arbitrary shape, taking into consideration the potential noise present in the data distribution (Figure 4). More details about the DB-SCAN algorithm applied to the gray level intensity profiles is reported in Appendix B. 

#### 2.4.2. ASM Model Application on Network Output

The obtained model described in Section 2.4.1 is subsequently applied on the vertices that are obtained in output from the 3D CNN (Section 2.3) after performing isotropic upsampling. The model is applied on the input volume at its original resolution (scale = 1). The input for the mean shape model application algorithm is the matrix of vertices that correspond to a single segmented volume (i.e., 3D CNN binary output). The normal directions to the surface are computed as described previously and a gray level intensity profile (*g*) is placed along each normal direction with a length $n=2*n_{s}+1$ where *n_s_* corresponds to the search length in both directions. PCA is applied to the matrix containing the selected gray level profiles to compute the eigenvector matrix $W_{g}$ and the mean gray level intensity profile $\bar{g}$. Noisy intensity profiles were filtered out before applying the PCA using the DB-SCAN clustering algorithm (see Appendix B). The objective function for the ASM evolution is defined as follows:(3)$$f=\sum \bar{bc}{\;}^{2}\;$$
where
(4)$$bc=W_{g}\prime *\left(g-\bar{g}\right)$$
(5)$$\bar{bc}=\frac{bc}{\sqrt{\lambda_{g}}}$$
and $\lambda_{g}$ are the eigenvectors. The regularization error is calculated as the sum of the objective function values for all the points corresponding to the search length within the profile length. The minimum value of *f* is selected at each iteration to determine the movement of each point as:(6)$$movement=\left(ind-1\right)-n_{s}$$
where $n_{s}$ is the length of the search profile in both directions as described previously and ind is the position in the profile at which *f* reaches the minimum value. The model is then realigned, removing the translation and rotation, using the same method as described in Section 2.4.1. The final *b* parameters are obtained as
(7)$$b=W_{s}\prime *\left(x_{search}-x_{mean}\right)$$
where $W_{s}^{\prime }$, *x_search_* and *x_mean_* are the shape eigenvectors matrix, the shape in the current iteration and the mean shape, respectively. The parameters are limited to the constraints of the mean model using the eigenvalues of the PCA model obtained during the training phase, as described previously. Hence, each point belonging to the final shape should lie within an area $b_{max}$ whose limits are given by a parameter *m*:(8)$$b_{max}=\pm m*\sqrt{\lambda_{s}}$$
where $\lambda_{s}$ are the shape eigenvalues. A detailed analysis of the shape constraint parameter (*m*) is given in Section 3. The *b* parameters of the model are then reset to the positions of the boundary in the original space:(9)$$x_{search}=x_{mean}+W_{s}*b$$

Finally, the translation and rotation are restored with an inverse aligning procedure (i.e., the inverse of the procedure described previously is applied).

#### 2.4.3. Post-Processing

The final vertices obtained in output from the active shape model need to be post-processed in order to compute validation parameters and evaluate the entire pipeline’s performance (Figure 5). While the final vertices are in theory very close to the real shape of the considered prostate, a post-processing algorithm is necessary to obtain the final binary segmentation of the volume. The algorithm is divided into 3 steps:
Triangulation: the Alpha Shape Triangulation method is employed to divide the 3D surface into a variable number of triangles, with α = 30.2D slices definition: to obtain the final 2D slices of the segmentation, the volume is divided into a number of planes whose z-coordinate corresponds to the number of slices. Then, similarly as to what was described previously, new vertices of the segmentation are found by computing the intersection between each ray and each triangle and taking the furthest point from the center, which is a first approximation of the points on the outermost surface.Final 3D volume reconstruction: the final 3D volume is obtained by stacking the 2D slices together, employing a hole-filling operation, and then a 3D morphological closing (spherical structural element, radius = 4). The post-processed binary volume is finally downsampled to the original resolution.

### 2.5. Performance Metrics

A comparison between manual and automatic masks is carried out to assess the performance of the proposed method for prostate gland segmentation. The Dice similarity coefficient (DSC) is calculated for both the train and test sets. Since prostate gland boundary delineation has a crucial role in radiation therapy planning, we also computed the Hausdorff distance. Specifically, we computed the 95th percentile Hausdorff distance (HD_95_), which is defined as the maximum distance of a set (manual boundary) to the nearest point in the other set (automatic boundary). This metric is more robust towards a very small subset of outliers because it is based on the calculation of the 95th percentile of distances. Finally, we calculated the relative volume difference (RVD) to measure the under- and over-segmentation of the algorithm [41].

## 3. Results

### 3.1. Ablation Study

We have tested our method in 4 different configurations by varying three of the most important parameters of the ASM: number of iterations (it), search length (ns), shape constraint (m). The *number of iterations* defines how many times the algorithm runs the search before stopping, the *search length* refers to the length in pixels along the profile normal to each surface point on which the search is performed, while the *shape constraint* is used to limit the model parameters based on what is considered a normal contour. Table 3 shows the ASM configuration used in this work, while Table 4 summarizes the performance of the proposed method in prostate segmentation.

The highest DSC value on the train set is obtained by the 3D network alone (0.886) while on the test set the best DSC is achieved by the VNet-T2 + ASM-2 configuration (0.851). In terms of Hausdorff distance, the best performing configuration is the VNet-T2 + ASM-2 both for the training and test set. The same configuration of ASM (it: 2, ns: 8, m:2) also obtains the best value of RVD in the test set, showing a good generalization ability of the proposed method. The HD_95_ analysis reveals a maximum distance between surfaces of about 7.55 mm, while RVD has an average value of 9.60%, meaning that the algorithm tends to over-segment on average. Interestingly, the Hausdorff distance always decreases with the application of ASM for all four tested configurations (both mean values and standard deviation). Furthermore, the application of the ASM model reduces the performance gap between the train and the test set, thus mitigating the overfitting of the VNet-T2 network. Figure 6 shows the performance of our method before and after the application of the ASM (i.e., VNet-T2 vs. VNet-T2 +ASM-2).

### 3.2. Inter-Observer Variability

In the research field of prostate gland segmentation, several studies have shown the presence of a fair amount of variability between the manual annotations of two operators [42,43]. Therefore, we asked a second experienced operator to manually segment the prostate volumes of the test set. Table 5 reports the metrics values (DSC, HD95, and RVD) comparing Operator 1 (Op1) vs. Operator 2 (Op2) to take account of inter-operator variability. Specifically, for the test set we compared the minimum, average and maximum values of the metrics obtained between two operators and the best performing configuration of our method (i.e., VNet-T2 + ASM-2). As can be seen from Table 5, the application of ASM improves the performance of the deep volumetric network, increases the average values of the metrics, reduces the outliers (maximum values) and presents values similar to those obtained between two operators (Op1 vs. Op2). Figure 7 depicts for three patients in the test set the comparison between VNet-T2 and the same network with ASM refinement.

## 4. Discussion

Prostate segmentation in MRI images is a challenging task, due to the heterogeneity in size, morphology, and appearance of the prostate gland. In addition, the boundaries between the prostate gland and surrounding anatomic structures are not always clear and defined. In this study, we propose a fully automated hybrid approach for 3D MRI segmentation of the prostate using CNN and ASM models. A 3D CNN, named VNet-T2, is employed to obtain an initial segmentation of the prostate gland. The output of the CNN is then used as the initialization of the statistical model (ASM) to obtain an accurate segmentation of the prostate contours. The main reason why we proposed a hybrid model based on CNN and ASM is that the CNN 3D model does not guarantee a noise-free prediction, which leads to some boundaries that deviate from the real ones. When applying ASM, the overall segmentation is constrained by the average shape model of the training set, and this ensures the plausibility of the shapes obtained as the final output, reducing the noise problem.

The results obtained show that the application of the statistical model in cascade to the CNN improves the segmentation performance, increasing the DSC values from 0.840 to 0.851 and decreasing the Hausdorff distance from 10.74 mm to 7.55 mm. These results show how applying the ASM on the CNN output results in a closer to true segmentation, especially at the contours. From Table 4, it can be seen that the ASM model allows us to decrease the standard deviation of the metrics, confirming the robustness of the proposed method. The use of a hybrid model allows us to overcome the limitations of the individual models, taking from each what are the most advantageous features. In fact, the overall segmentation can be improved by incorporating knowledge of prostate shape variability into a CNN-based framework.

Despite the good performance, this method is not free from limitations. The CNN architecture presented here is quite simple, and to achieve better results, it would be necessary to increase the depth of the network and the number of base filters. This is because the ASM model performs better the more precise its initialization is. Currently, it was not possible to further increase the depth of the network without running into the problem of overfitting. This problem may be solved in the future by increasing the size of the dataset. This limitation can be partially overcome by adding terms that better approximate the shape of the CNN output by penalizing the curvature within the triangulation algorithm or by using another triangulation algorithms [44]. Moreover, since the ASM was developed for operating on T2w MRI images, if the MRI modality were to change (e.g., ADC, T1w, etc.), the model would have to be partially fine-tuned again. The constructed mean shape model could be maintained since it depends on the organ and not on the image acquisition modality, but the grayscale appearance model would have to undergo another fine-tuning process.

In the future, we plan to expand the dataset, including volumes acquired using different scanners so as to increase the capability of generalizing the problem both for the deep learning network and the ASM. The presented pipeline is also modular, so that in the future it would be possible to update or implement newly developed deep neural networks and/or active shape model. Our methodology (i.e., integration of an ASM with a 3D CNN) can be easily extended to other segmentation problems, including the segmentation of the other organs (e.g., lung, muscles, etc.) and also segmentation within images acquired using a different imaging modality (e.g., CT, US). Our research group is currently working on the extension of this approach for prostate gland segmentation in multiparametric magnetic resonance imaging (mp-MRI), with the aim of developing innovative computer-aided diagnosis systems for prostate cancer characterization.

## 5. Conclusions

In this paper, we propose a fully automated hybrid approach for 3D MRI prostate segmentation using CNN and ASM models. The combination of semantic features extracted from the CNN and statistical features from the ASM model results in performance comparable to that of manual operators. The most important finding of this work is that the use of a CNN-based initialization for an ASM model improves segmentation performance compared to the deep network alone. The quantitative differences between this hybrid approach and the 3D network may not be large, but the improved accuracy could prove to be critical for specific clinical applications, allowing the conservation of smaller structures such as the bladder, external sphincter, or seminal vesicles.

The publicly available code can be used to compare the proposed method against popular state-of-the-art models and newly developed architectures. Given the direct clinical necessity of prostate segmentation and volumetric analysis for surgical planning and fusion biopsy, our approach could represent a promising model for future clinical implementations of deep learning to clinical imaging.

## Figures and Tables

**Figure 1 jimaging-08-00133-f001:**
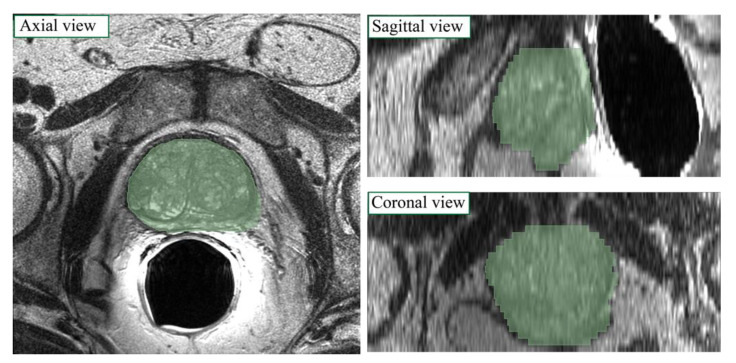
Manual label superimposed on MRI image in axial, sagittal and coronal views for a sample patient.

**Figure 2 jimaging-08-00133-f002:**
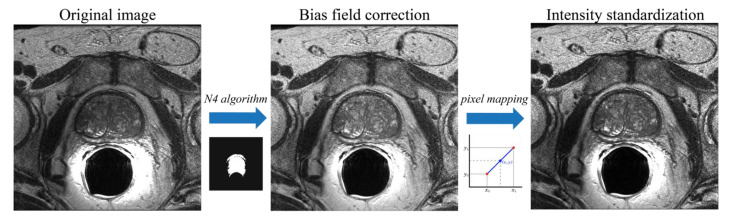
Pre-processing steps applied to each MRI volume. First, the N4 algorithm is used for bias field correction. Then, intensity normalization is applied to standardize each MRI volume.

**Figure 3 jimaging-08-00133-f003:**
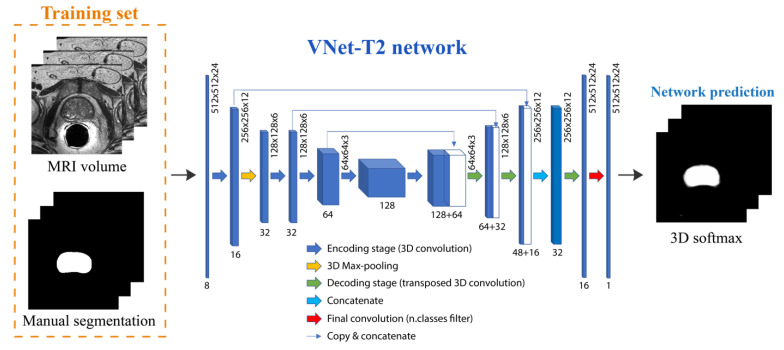
Architecture of the deep network employed in this work. Starting from the 3D MRI volume, the VNet-T2 network performs a volumetric segmentation of the prostate gland.

**Figure 4 jimaging-08-00133-f004:**
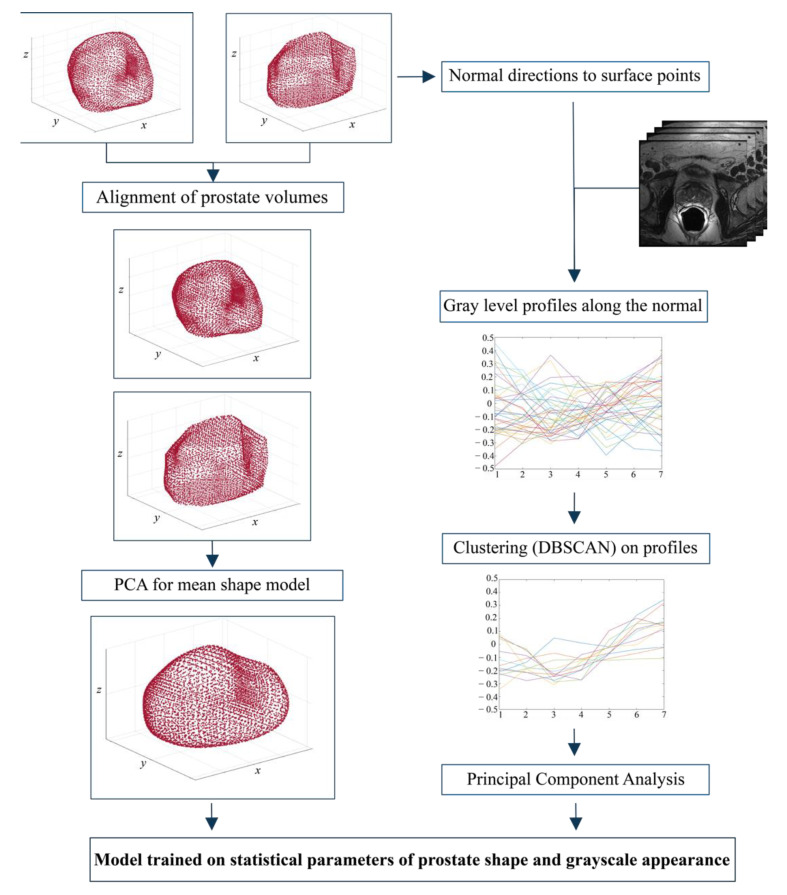
Steps followed to create (i) the average prostate shape model and (ii) the appearance of gray levels used to optimize the prostate contour. The mean shape model is calculated by applying the principal component analysis (PCA) after realigning all volumes in the training set (36 patients). On the other hand, the gray level profiles of the original images are used to construct the grayscale appearance model.

**Figure 5 jimaging-08-00133-f005:**
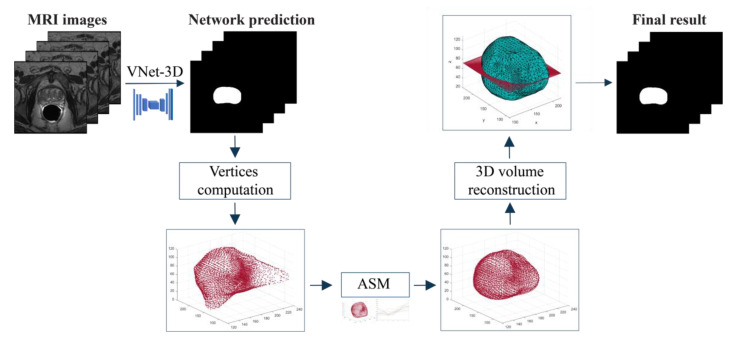
Schematic representation of the proposed algorithm. A first segmentation is provided by the VNet-T2. Then, the ASM model is applied to refine the volumetric segmentation. Finally, triangularization is performed to obtain the binary masks.

**Figure 6 jimaging-08-00133-f006:**
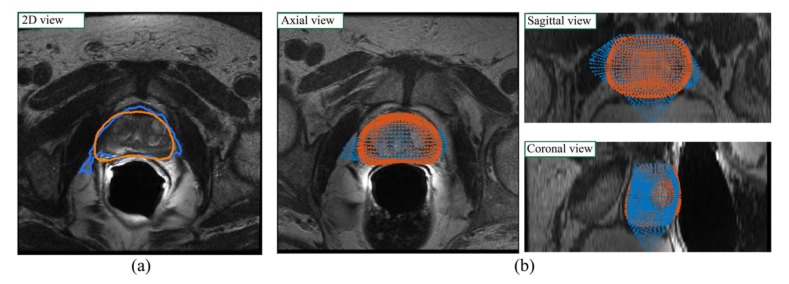
Visual performance of the proposed method before (blue) and after (red) the ASM model. The blue contours represent the output of the VNet-T2 network while the orange contour is the result obtained with the combination of the VNet-T2 network and the ASM model (VNet-T2 + ASM-2). (**a**) 2D view; (**b**) 3D view in the axial, sagittal, and coronal planes. The introduction of the active shape model to refine the prostate contour increased the accuracy of the gland segmentation especially in the base and apex zones.

**Figure 7 jimaging-08-00133-f007:**
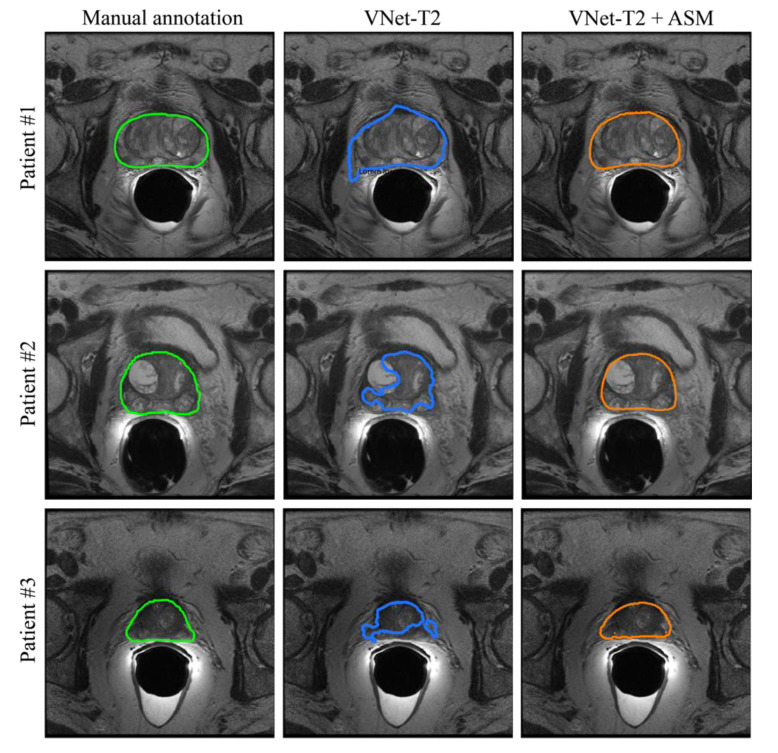
Comparison between manual annotations (first column) and the segmentation obtained for three patients of the test set. The second column shows the results obtained with only the application of the 3D network (VNet-T2) while the ASM refining (VNet-T2 + ASM-2) is illustrated in the last column. Prostate segmentation can be improved by incorporating knowledge of prostate shape variability (ASM) with a deep network prediction.

**Table 1 jimaging-08-00133-t001:** Previously published methods for prostate segmentation in MR images. The table presents the problem addressed for each method, along with details of the datasets and the proposed solutions.

Reference	Year	Dataset	Problem	Solution
Cheng et al. [15]	2016	100 axial MR images	Image artifacts; Large inter-patient shape and texture variability; Unclear boundaries	Atlas-based model combined with a CNN to refine prostate boundaries
Yu et al. [12]	2017	80 T2w images	Limited training data	Volumetric ConvNet with mixed residual connections
He et al. [6]	2017	50 T2w axial MR images	Variability in prostate shape and appearance among different parts and subjects	Combine an adaptive feature learning probability boosting tree with CNN and ASM
Kamiri et al. [16]	2018	49 T2w axial MR images	Variability of prostate shape and appearance; small amount of training data	Stage-wise training strategy with an ASM embedded into the last layer of a CNN to predict surface keypoints
Zhu et al. [13]	2020	50 T2w images	Prostate variability; Weak contours; Limited training data	Boundary-weighted domain adaptive neural network
Jia et al. [14]	2020	80 T2w images	Anisotropic spatial resolution	As-Conv block: two anisotropic convolutions for x-y features and z features independently
Ushinsky et al. [17]	2021	299 T2w images	Variability in prostate appearance among different subjects	Customized hybrid 3D-2D U-Net CNN architecture
Meyer et al. [18]	2021	89 T2w images	Anisotropic spatial resolution	Fusion of the information from anisotropic images to avoid resampling to isotropic voxels.
Pollastri et al. [19]	2022	Prostate-MRI-US-Biopsy dataset [21,22,23]	Variability of prostate shape and texture	Long-range 3D Self-Attention Block integrated within the CNN

**Table 2 jimaging-08-00133-t002:** Hyperparameters of the VNet-T2 network.

Hyperparameter	Chosen Value
Network depth	4
Number of base filters	8
Number of trainable parameters	1.192.593
Learning rate	10^−4^
Loss function	Dice similarity loss
Metric	Dice score

**Table 3 jimaging-08-00133-t003:** Configurations of the four ASM tested in this work.

Name	Number of Iterations (it)	Search Length (ns)	Shape Constraint (m)
ASM-1	1	8	3
AMS-2	2	8	1
ASM-3	2	8	2
ASM-4	2	8	3

**Table 4 jimaging-08-00133-t004:** Performance of the proposed strategy on train, validation and test sets. VNet-3D indicates the result of the segmentation by adopting only the 3D network described in Section 2.3. VNet-3D + ASM indicates the combination of the 3D network with the 4 configurations of the ASM. Best values are highlighted in bold. HD_95_: 95th percentile Hausdorff distance; RVD: relative volume difference.

Method	Subset	DSC	HD_95_ (mm)	RVD (%)
VNet-T2	Train	**0.893 ± 0.020**	7.94 ± 3.16	**8.58 ± 5.52**
	Val	0.851 ± 0.027	6.98 ± 1.55	11.65 ± 7.01
	Test	0.840 ± 0.039	10.74 ± 5.21	11.22 ± 7.85
VNet-T2 + ASM-1	Train	0.880 ± 0.033	6.79 ± 3.06	11.92 ± 7.58
	Val	0.858 ± 0.028	6.89 ± 1.89	9.78 ± 4.86
	Test	0.839 ± 0.055	8.87 ± 3.39	12.87 ± 4.53
VNet-T2 + ASM-2	Train	0.870 ± 0.039	**6.05 ± 1.92**	9.45 ± 8.53
	Val	**0.859 ± 0.042**	**6.44 ± 2.08**	9.58 ± 9.92
	Test	**0.851 ± 0.044**	7.55 ± 2.76	**9.60 ± 7.80**
VNet-T2 + ASM-3	Train	0.878 ± 0.035	6.87 ± 3.47	9.38 ± 7.88
	Val	0.853 ± 0.038	5.82 ± 1.05	**8.09 ± 5.91**
	Test	0.842 ± 0.049	7.26 ± 2.69	11.63 ± 9.31
VNet-T2 + ASM-4	Train	0.877 ± 0.036	6.73 ± 3.26	10.05 ± 7.92
	Val	0.851 ± 0.038	6.48 ± 1.36	9.75 ± 5.87
	Test	0.839 ± 0.052	**7.40 ± 2.79**	12.72 ± 9.99

**Table 5 jimaging-08-00133-t005:** Minimum, mean, and maximum values of metrics in the test set compared with inter-operator variability (Op1 vs. Op2).

Method	DSC			HD_95_ (mm)			RVD (%)		
	Min	Average	Max	Min	Average	Max	Min	Average	Max
Op1 vs. Op2	0.842	0.892	0.935	2.57	4.51	8.64	1.21	15.90	25.38
VNet-T2	0.783	0.840	0.908	5.00	10.74	22.89	1.79	11.22	29.22
VNet-T2 + ASM-2	0.761	0.851	0.917	3.80	7.55	12.78	0.33	9.60	27.87

## Data Availability

The image dataset and the scripts used to develop the VNet-T2 for prostate segmentation in MR images can be found here: 10.17632/25r9ccd7z2.1 (doi).

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
