# Peer review of "Integration of Deep Learning and Active Shape Models for More Accurate Prostate Segmentation in 3D MR Images"

_2313-433X, 2022, doi:10.3390/jimaging8050133_

Round 1

Reviewer 1 Report

  • The abstract should be improved. Why does the automatic segmentation of the prostate allow to bypass the huge workload of
    technicians? Do you mean physicians?
  • Automatic segmentation does not allow to “obtain operator-independent results.” Unless the model is trained using a dataset segmented by several physicians from different centers using multicenter/multivendor dataset. Please correct.
  • In section 2.3 “it was decided to avoid resampling to make the sizes compatible” Can the authors comment more on this ? since resampling usually allows to have an isotropic resolution to remove variance in scanner resolution.
  • There is no new architecture in the work, it is a U-Net. The name of the model is confusing.
  • What are the results without class balancing
  • In section 2.1 “Finally, the dataset was divided into a training set (45 patients) and test set (15 patients).” It is unclear to me if the authors used data among the 45 patients for validation. Are the 15 patients used as a holdout data and tested on only when the training is complete or as a validation data? How did the authors choose the best model? Authors should clarify this point and discuss the impact of the ratio of the 3 sets (Training/validation/Testing) on the statistical power of the results
  • Please include the learning curve
  • How was overfitting ruled out ?
  • Minors:
  • Please add a take away message for the figures
  • The number are hard to read in Figure 4

Reviewer 2 Report

The conclusion part should be improved. Please discuss the limitation of this research. For manual segmentation just there was one expert and we see a lack of inter and intraobserver study.

Reviewer 3 Report

The Authors proposed deep learning-based prostate cancer method. Overall paper is well written and interesting. I have the following concerns:

1) In Abstract authors missed the information, that how existing methods are lacking in the same topic

2) In my opinion the Introduction is lacking with recent 2021-2022 literature.

3) Picking some method is not a good way, authors need to explain why they choose the U-Net like structure ? Where the multiscale structures like DeepLab V3 are available 

4) Author needs to provide a difference Table including the differences of Proposed VNet-T2 , V-Net and U-Net

5) I would suggest making a section for Ablation study with the same experiments/ new experiments

6) The results are good, but the Conclusion in the current form is not acceptable, Author needs to reflect on the claims of the Abstract and future directions.

7) Include limitations of the proposed method

Round 2

Reviewer 1 Report

The authors have addressed most of my concerns. Thank you.

Reviewer 3 Report

The authors correctly responded to the review questions. I vote for the publication of this paper in its current form.